# In Silico Logical Modelling to Uncover Cooperative Interactions in Cancer

**DOI:** 10.3390/ijms22094897

**Published:** 2021-05-05

**Authors:** Gianluca Selvaggio, Claudine Chaouiya, Florence Janody

**Affiliations:** 1Fondazione the Microsoft Research—University of Trento Centre for Computational and Systems Biology (COSBI), Piazza Manifattura 1, 38068 Rovereto, Italy; selvaggio@cosbi.eu; 2Instituto Gulbenkian de Ciência, Rua da Quinta Grande 6, 2780-156 Oeiras, Portugal; 3CNRS, Centrale Marseille, I2M, Aix Marseille University, 13397 Marseille, France; 4i3S—Instituto de Investigação e Inovação em Saúde, Universidade do Porto, Rua Alfredo Allen, 208, 4200-135 Porto, Portugal; 5IPATIMUP—Instituto de Patologia e Imunologia Molecular da Universidade do Porto, Rua Dr. Roberto Frias s/n, 4200-465 Porto, Portugal

**Keywords:** cooperative oncogenesis, logical computational model, signalling pathways, tumour microenvironment, epithelium to mesenchymal transition

## Abstract

The multistep development of cancer involves the cooperation between multiple molecular lesions, as well as complex interactions between cancer cells and the surrounding tumour microenvironment. The search for these synergistic interactions using experimental models made tremendous contributions to our understanding of oncogenesis. Yet, these approaches remain labour-intensive and challenging. To tackle such a hurdle, an integrative, multidisciplinary effort is required. In this article, we highlight the use of logical computational models, combined with experimental validations, as an effective approach to identify cooperative mechanisms and therapeutic strategies in the context of cancer biology. In silico models overcome limitations of reductionist approaches by capturing tumour complexity and by generating powerful testable hypotheses. We review representative examples of logical models reported in the literature and their validation. We then provide further analyses of our logical model of Epithelium to Mesenchymal Transition (EMT), searching for additional cooperative interactions involving inputs from the tumour microenvironment and gain of function mutations in NOTCH.

## 1. Oncogenesis: A Dynamic Process of Multifactorial Nature

### 1.1. Cooperative Oncogenesis

The development of omics technologies has permitted the identification of numerous mutations in tumours across a large number of cancer types, thousands of genes upregulated or downregulated, as well as altered epigenetic modifications [1]. Some of these alterations are drivers of oncogenesis. They have been classified as oncogenes or tumour suppressors, depending on whether they are constitutively activated by gain-of-function (GoF) mutations or inactivated by loss-of function (LoF) mutations, respectively. In contrast, other alterations are likely non-oncogenic with no selective advantage. The search for critical drivers of oncogenesis revealed that only a small number of solid tumours arise from mutations in single loci [2]. Most often, a sequence of randomly occurring mutations in oncogenes or tumour-suppressor genes is required for the series of events leading to traits associated with malignant cancer phenotypes [3]. These genomic alterations disrupt the functioning of intertwined signalling networks [4]. Estimates of the number of mutations causing the large majority of malignant cancer in human vary from 3 to 12, depending on the cancer type, and could be even larger in organs with rapid turnover [5]. According to this multiple-hit hypothesis, in vitro experiments using cell lines, as well as in vivo models of cancer, such as those established in *Drosophila*, confirmed that at least two cooperating mutations are required to initiate oncogenesis [6,7,8,9,10]. In this article, we define how two factors cooperate when the presence (or absence) of both is required to produce a given outcome.

In addition to intracellular cooperative mechanisms, the success or failure of oncogenesis depends on interactions between tumour cells and the Tumour Microenvironment (TME), which includes the Extracellular Matrix (ECM), as well as non-cancerous stromal cells present within, or adjacent to, the tumour. While a healthy state of the microenvironment can help protect against tumorigenesis and invasion, a non-healthy microenvironment, reshaped by cancer cells, becomes an accomplice to tumour cells supporting their survival, invasive, and metastatic abilities [11]. Cooperative behaviours are also observed between groups (clones) of tumour cells carrying different genetic alterations, including evidence supporting the requirement of interactions between distinct clones for tumour progression [12,13]. In addition, clonal cooperativity has been reported between cancer cells with no discernible genetic differences to influence their aggressiveness, suggesting a key role of epigenetic modifications [14,15]. Mechanisms of cooperativity in play likely involve paracrine signalling between clones of cells [12,14].

Thus, while tumour progression is dictated by the cooperative action of multiple intracellular molecular alterations, it also depends on the synergy between cancer cells and between cancer cells and the TME. Therefore, identifying the cooperative determinants responsible for the emergence of cancer phenotypes is of the utmost relevance for developing efficient therapeutic approaches. Still, the search for these factors faces major challenges. Even though some oncogenes and tumour-suppressors may be more frequently mutated or commonly deregulated in some carcinomas, tumours display very distinct patterns of mutations and signalling activities, even within the same tumour type [16]. Moreover, altered activity of a signalling pathway does not necessarily reflect mutations in genes encoding for components of that pathway [17]. Moreover, many genes can act as either oncogenes or tumour suppressor genes in different experimental settings. The opposite effects of these “chameleon” genes depend on signalling networks whose activity is defined by the genomic context (different genetic background) and by the state of the microenvironment [18]. Thus, cooperative interactions are likely highly dynamic during tumour evolution, owing to the emergence of novel mutations, epigenetic alterations, and signals present in the TME.

### 1.2. Experimental Assessment of Cooperative Oncogenesis and Current Limitations

Providing evidence of cooperativity between two molecular alterations, or between two cell populations, requires that the phenotypic outcomes induced by the presence of both differ from the behaviour produced by each one alone. Established cancer cell lines have been valuable models to determine the role of particular molecular lesions in dictating the malignant cancer phenotypes [19]. However, the large majority of these in vitro models fail to recapitulate the multistep development of cancer initiation and progression. Moreover, despite extensive technological developments, the complex interactions between cancer cells and the TME can hardly be fully reconstituted. Circumventing some of these difficulties, experimental genetic models have been developed to assess the interplay between distinct molecular lesions and between cancer cells and the TME during cancer progression. Among those, diverse mice models have been generated via a variety of methods, including chemical or physical mutagenesis (gamma rays, X-rays, UV-light, and particle radiation), viral infection, insertion of transgenes, homologous recombination, and, more recently, genome editing by the CRISPR technology [20]. These models can mirror complex mutational patterns observed in tumour samples, as well as interactions taking place within a microenvironment. However, these approaches remain time-consuming and costly, mostly calling for highly trained personnel and advanced infrastructure. In contrast, the fruit fly *Drosophila melanogaster* is a cheap in vivo model, which can be genetically modified and maintained with comparatively basic training and infrastructure. It displays high levels of conservation in core cancer relevant genes and signalling pathways. It supports easy and fast generations of lines carrying multiple mutated transgenes recapitulating mutational patterns observed in cancer samples as well as genetic modifier screens [10,21]. In addition, genetic tools developed in *Drosophila* allow assessment of how genetically engineered cancer cells interact with their wild type neighbours or cooperate for their survival and expansion [10]. Yet, as the fly lacks similarities in telomere and telomeric maintenance strategies compared to human and do not possess fibroblasts, adaptive immune systems, and vasculature, the role of these players can hardly be evaluated in *Drosophila* cancer models.

Research using these experimental models made tremendous contributions to our understanding of oncogenesis. Yet, these approaches remain labour-intensive and challenging to accurately anticipate causative combinations of molecular lesions and microenvironment interactions in tumour initiation and cancer progression. To overcome such a hurdle, an integrative, multidisciplinary effort is required, combining assets of different approaches to characterise the interplay of multiple regulatory components in cancer complexity.

## 2. Computational Modelling Approaches to Pinpoint Cooperative Interactions in Oncogenesis

Computational models can simulate the complex dynamics driven by intricate signalling pathways with feedbacks and cross-talks. As such, they are effective tools to help decipher cooperative mechanisms in the context of cancer biology. To explain cancer phenotypes at the molecular level, these models integrate interactions documented in the literature and data generated using high throughput omics technologies (e.g., genomics, epigenomics, transcriptomics, metabolomics, proteomics, and others). Different formalisms, from systems of differential equations to logical models, can be used to portray molecular interactions [22,23,24]. In this review, we focus on discrete, logical models.

### 2.1. Basics of Logical Modelling

Unlike differential equations, logical models do not require precise values for the concentrations of molecular species, gene expression levels, or kinetic constants that are mostly lacking in the literature. Logical models allow the integration of a variety of regulatory and signalling interactions between a relatively large number of components. In addition, these models support non-deterministic asynchronous dynamics, making them biologically more plausible than deterministic trajectories.

Briefly, logical models include collections of nodes (or variables), which can represent almost anything, including gene activity, presence, activity of a protein, or the state of a cell. Each node is associated with a discrete variable, which is a logical (often Boolean i.e., binary) abstraction of its level of activity (1 for active, 0 for non-active). Yet, these models handle multi-valued variables, which are critical to convey distinct thresholds to different functional effects. Nodes are linked to each other by signed arcs (arrows connecting variables), which indicate regulatory relationships between these variables (i.e., inhibitions or activations). Figure 1A illustrates a regulatory graph associated to a simple Boolean model. Finally, the value of each variable is defined by a logical regulatory rule, which depends on the variables influencing that variable, i.e., its regulators (Figure 1B). Input nodes that embody external signals are maintained constants (this is the case of the node S in the Figure 1). The values of the model variables define the model states, and the model dynamics are illustrated by the sequences of states, which can be represented by the so-called state transition graphs (Figure 1C). Given a state, its successors are determined by the logical rules. Some variables are called to change their values due to their regulators, while some remain stable. The asynchronous updating mode performs these updates independently, leading to non-deterministic dynamics (i.e., alternative, concurrent trajectories). As the number of states is finite, model dynamics are necessarily trapped into attractors, which are either stable states (states in which all the variables are stable), or complex attractors (sets of states in which the model dynamics evolves indefinitely, denoting an oscillatory behaviour). Properties of interest mostly relate to the model attractors: their characterisation and their reachability from given initial conditions. For example, with the simple model of Figure 1, there are two stable states occurring when the input node is set to value 1. They are both reachable from some states, and one can evaluate the reachability probability of each. To do so, a Monte Carlo approach, consisting of N (asynchronous) simulation runs starting from an initial state and counting the number R of runs ending in e.g., the stable state (0111), provides R/N as a good approximation of this probability [25]. Finally, the logical modelling enables an easy assessment of the impact of perturbations, where LoF or GoF are defined by blocking the value of the corresponding variable to 0 or 1, respectively. Figure 1D provides an illustration of a perturbation that leads to the loss of a stable state. We refer to [26] and references therein for further detail on the logical formalism

### 2.2. Logical Models and Their Contribution to Cooperative Oncogenesis

Since Kauffman’s seminal work, Boolean models and their attractors, embodying cellular phenotypes, have been considered relevant to explore tumorigenesis [27,28]. Whereas early Boolean models were conceptual representations of genetic networks, the remarkable development of omics technologies allowed the identification of interaction networks and the construction of effective, logical models. Meanwhile, the modelling formalism evolved to better account for the behaviour of regulatory networks [26,29,30]. Computational methods have been developed to map omics data to models, or to train logical models on data, thus enabling model contextualisation [31,32,33,34]. Here, we review representative examples of logical models reported in the literature that supported the identification of cooperative mechanisms in oncogenesis. This overview also illustrates the lively activity of the research community involved in logical modelling.

Basically, the procedure to uncover cooperative mechanisms between X and Y from a computational model consists of comparing the outcomes of that model in three conditions: when X alone is altered, when Y alone is altered, and when both are altered. Novel outcomes in the third condition reveal cooperativity. Altering a model component entails fixing the value of the corresponding variable, conveying the presence of a signal or the mutation of the corresponding molecular component.

Several modelling works focused on networks controlling cell proliferation and cell death, as dysregulated balance between these processes is a well-known hallmark of cancer. Grieco et al. proposed a logical model assessing the involvement of specific components of the MAPK signalling network in controlling cell fate decisions between proliferation, growth arrest, and apoptosis in the context of bladder cancer [35]. Exploring the impact of different combinations of external signals and of network perturbations, their model predicts a cooperation between EGFR overexpression and the loss of p53 or of p14 in inducing proliferation. In contrast, an overexpression of EGFR or an FGFR3 activating mutation, combined with a stimulus of the TGFB receptor, would lead to growth arrest and apoptosis. Still, regarding bladder cancer, and considering the same cellular responses, a logical model focusing on the E2F pathway was developed by Remy et al. to explain patterns of genetic alterations (co-occurrences and mutual exclusivities) observed in tumour data [36]. The authors identified diverse. predicted cooperative genetic interactions supporting tumour aggressiveness or invasion. For instance, since FGFR3 and PI3K mutations co-occur in the literature, the authors explored the difference in proliferation for the three mutants in the model: overexpression of PI3K alone, of FGFR3 alone and of both PI3K and FGFR3. They did not find cooperation between both mutants in uncontrolled growth. Instead, it appeared that a third mutation, the deletion of CDKN2A, was necessary to unquestionably increase proliferation. Accordingly, co-occurring FGFR3 activating mutations and CDKN2A deletions have been reported in the literature and several datasets. In addition, the model of Remy et al. suggests that, to obtain a significant increase in proliferation in the context of a FGFR3 GoF and CDKN2A LoF mutations, a third (activating) mutation in p21CIP is required. These predictions were supported by datasets from the TCGA database, which provides genomic data from over 20.000 primary cancer and matched normal samples. Among the five tumours carrying mutations in FGFR3 and p21CIP, four also display a homozygous deletion of the CDKN2A gene. With this model, the authors could also predict the temporal order of gene alterations favouring proliferation. For instance, the co-occurrence of E2F2 amplification and p53 LoF would be beneficial for the tumour cell when mutations in *p53* appear first. More recently, Rossato et al. built a Boolean model of TGFB signalling to identify cooperating mutations that would explain the dual role of TGFB signalling in promoting cell cycle arrest or apoptosis at early stages of cancer, but proliferation at later stages [37]. Model simulations indicate that simultaneous LoF of SMAD2/3, p38MAPK and p53 are required to trigger a proliferative phenotype.

Another hallmark of tumour cells that has motivated a substantial number of logical models is the Epithelial to Mesenchymal Transition (EMT), a representative example of cancer cell plasticity [38]. This process not only involves a switch between the two extreme phenotypes, epithelial and mesenchymal, but also the transition to a spectrum of incomplete EMT phenotypes. These hybrid phenotypes, which co-express epithelial and mesenchymal markers, have been proposed to provide pluripotent abilities to cancer cells, resistance to chemotherapeutic drugs, and increased aggressive potential [39]. A model of EMT, driven by TGFB, was published by Steinway et al. [40]. Model prediction and experimental validation in hepatocellular carcinoma cell lines indicates that the concomitant activation of the WNT and SHH signalling are required to induce a mesenchymal phenotype downstream of TGFB. Relying on this model, a control network strategy was then used to screen single and multiple perturbations suppressing TGFB-driven EMT. The authors observed that single perturbations may favour hybrid EMT phenotypes, while multiple interventions, involving SMAD inhibition, restore an epithelial phenotype. Assessment of model predictions in HCC cell lines, confirmed that a combined targeting strategy could prevent ECad expression and cell migration. Combinatorial interventions would thus be required to fully suppress invasive properties of tumour cells [41].

Analogously, Cohen et al. built a logical model of cancer to investigate EMT sensitivity to different perturbations and validated it against lung cancer transcriptomics data [42]. Single LoF or GoF and double mutants were systematically generated and analysed to identify mutant combinations altering the probability of reaching a metastatic phenotype. From this analysis, a synergy emerged between NOTCH GoF and p53 LoF, which, together, lead to metastatic phenotype in 100% of the runs, whereas the probability to reach this phenotype for the single Notch GoF and for the p53 LoF is below 36%. Méndez-López et al. presented a Boolean model, demonstrating that the dynamic interplay between nine key regulatory components can explain the temporal sequence by which epithelial cells transit through a senescent state to acquire a mesenchymal stem-like phenotype [43]. In agreement with experimental data reported in the literature, this model indicates that cellular inflammation, simulated by the constant overactivation of NFKB, results in a higher likelihood of reaching a mesenchymal stem-like state. Still focusing on EMT, and using cell adhesion properties as read-outs for the acquisition of EMT phenotypes, we recently published a model, which accounts for the epithelial, hybrid, and mesenchymal phenotypes acquired by cancer cells [44]. Main outcomes of this model, and novel analyses that lead to the identification of cooperative mechanisms underlying the EMT process, are described in the next section.

Computational models also allow exploring drug effects, predicting therapeutic strategies [32], and the references therein. For example, Flobak et al. defined a logical model of the cellular network controlling growth in gastric cancer, including two output nodes, anti-survival and pro-survival, serving as phenotypic read-outs [45]. Searching for pairs of inhibitors of cell growth, the model predicted five synergistic inhibitions involving seven components. Among those, four cooperative inhibitions were confirmed experimentally, by comparing the effect on cell growth of treating the AGS gastric cancer cell line with chemical inhibitors of the seven proteins, in single or combinatorial formulations, using cell growth real-time assays. To identify targeted combinatorial therapies in colorectal cancer (CRC), Eduati et al. looked for dynamic interactions between different signalling pathways and cell-specific drug responses [31]. By measuring a set of phosphoproteins under different combinations of stimuli and inhibitors, they could instantiate cell line-specific logical models of underlying signalling networks, reflecting the heterogeneity of 14 CRC cell lines. This study shows that, unlike genomic features, which have limited predictive power of sensitivity to kinase inhibitors, the dynamics of signalling pathways can determine the efficacy of targeted drug treatments. In particular, the authors could validate the combined blockade of MEK and GSK3 as a strategy to overcome resistance to MEK inhibitors, a scenario that could not be found based on associations with genomic data.

So far, most models have been defined on the basis of extensive literature reviews, and/or from information on interactions stored in dedicated databases. These models are somewhat “generic,” as they intend to represent an “average cell,” and thus do not account for tumours and patient heterogeneity. Hence, software tools, such as CellNOptR [34,46] and PRUNET [47], have been defined to contextualise models using high-throughput data, leading to cell type specific models (Ref. [47] provides an overview of these tools). To personalise logical models, Béal et al. recently proposed an approach integrating mutation data, copy number alterations (CNA), transcriptomic, and proteomic data to models [33]. The authors illustrated the value of their framework using a generic model of cancer signalling pathways [48] and breast cancer data from the METABRIC project, including RNA expression data, mutation profiles, CNA and clinical data [49,50]. As illustrated in Figure 1, of [33] data need to be appropriately processed through functional inference (in the case of discrete genomic data, such as mutations or CNA) and through discretisation and normalisation (in the case of e.g., expression data).

Finally, it is important to note the coordinated effort in the computational biology community to establish a standard format to store and exchange models [51], to define computational methods handling ever larger models, as well as to develop several software tools available to the community (e.g., see [26]). The Consortium for Logical Model and Tools aims to promote these developments [52]. As an illustration of these activities, the CoLoMoTo notebook provides a computational framework combining several tools [53]. A tutorial presenting a workflow to predict mutations enabling tumour cell invasion from a specific model illustrates the use of this notebook [54].

## 3. Logical Modelling Predicts Cooperative Signals Governing Phenotypes Amid the EMT Continuum

Using a logical model of the EMT cellular network, we have identified cooperative signals controlling cancer-associated phenotypes amid the EMT continuum [44]. This in silico model encompasses 39 intracellular nodes, including EMT transcription factors, epithelial (ECad and miR200) and mesenchymal (SNAIL, SLUG, ZEB, TCF/LEF, BCat) markers and known EMT signalling pathways (RAS, NOTCH, WNT, TGFB, JAK/STAT, Hippo, Integrins and AKT). These components are controlled by 10 inputs from the tumour microenvironment (HGF, EGF, ECM stiffness, TGFB, IL6, DELTA, ROS, WNT and the ligands for RPTP and FAT4). The model includes 2 read-outs typically affected during EMT, notably Adherens Junctions (AJs) stability and Focal Adhesions (FAs) remodelling (Figure 2). It robustly recapitulates the phenotypic diversity observed in the EMT continuum, including pure epithelial (E) and mesenchymal (M) phenotypes, as well as 6 incomplete EMT phenotypes, among which are 3 hybrid (H) ones that co-express epithelial and mesenchymal markers (Table 1). Model predictions indicate that the FAK-SRC complex cooperates with a stiff ECM to upregulate SNAIL and to induce a full mesenchymal phenotype. Experimental validations using the MCF10A and MDCK cell lines with conditional SRC activation, grown on collagen gels of different Young’s moduli, which mirror a soft ECM surrounding normal mammary epithelial cells or a stiffer matrix reported for stroma adjacent to transformed cells, confirmed that SNAIL expression was significantly higher in SRC overactivating cells grown on stiff gels compared to those plated on soft ones. Moreover, while SRC overactivating cells grown on stiff gels were isolated and accumulated poorly ECad at the cell membrane, those plated on soft gels maintained epithelial features characterised by the presence of cell-cell contacts at ‘tip-like” junctions and membrane-associated ECad. In addition, model simulations revealed that FAK-SRC cooperates with RPTP, which mediates homophilic cell-cell adhesion, to gain a hybrid H3 phenotype, reminiscent to the one displayed by cancer cells that migrate collectively [39]. According to this prediction, forcing PTPR-kappa expression in MCF10A with conditional SRC activation using the (CRISPR)-based activation system could significantly restore cell aggregation. Thus, our model can uncover cooperative mechanisms leading to the acquisition of EMT phenotypes [44].

### 3.1. RPTP_L Cooperates with Inputs from the Microenvironment in EMT

To further extend our search for cooperative interactions involved in EMT dynamics, we analysed the phenotypes displayed by the model when switching single inputs or combinations of multiple inputs using GINsim (version 3.0, http://ginsim.org, accessed date 2 May 2021), a software dedicated to the definition and analysis of logical models [55]. When several phenotypes arise, we performed simulations starting from an epithelial state (E1) to evaluate the reachability probability of each, using the GINsim built-in functionality called Avatar, which performs a modified Monte Carlo simulation [25]. To ensure the convergence of estimated probabilities, 10^5^ runs were performed.

The underlying biological assumptions of our model were that, in the context of a non-tumorigenic TME, the ECM remains soft, growth factors (HGF and EGF), inflammatory signals (IL6, TGFB and ROS), DELTA and WNT are absent, whereas the RPTP ligand (RPTP_L) is present, as RPTPs display tumour suppressive capabilities [56]. Accordingly, when all the input nodes are set to 0, except RPTP_L that is fixed to 1, the epithelial phenotype E1 is stable (Figure 3A). We have previously reported synergistic effects involving RPTP in EMT [44]. Since fixing all inputs to 0, including RPTP_L, does not permit to leave the E1 phenotype, we searched for cooperative effects between the loss of RPTP_L and other inputs from the TME.

When fixing DELTA or WNT to 1 in the presence of RPTP_L, the model retains the E1 phenotype. Accordingly, mouse prostate cells expressing a constitutive active form of the DELTA receptor NOTCH are unable to metastasise [57]. The loss of RPTP_L is not sufficient to induce EMT in the presence of DELTA or WNT, as with DELTA or WNT set to 1, the model maintains an E1 phenotype in the absence of RPTP_L (Appendix A).

The model predicts that ROS and IL6 are sufficient to induce EMT phenotypes. It reaches a unique M2 mesenchymal phenotype in the presence of ROS (Figure 3A), mirroring the invasive ability observed when epithelial cells are exposed to oxidative conditions [58] and the M1 mesenchymal and H1 hybrid phenotypes in about the same proportions with IL6 sets up at 1 (Figure 3B). We do not observe synergistic effects between the loss of RPTP_L and ROS or IL6, as identical phenotypes were reached by fixing these inputs to 1 in the presence, or in the absence, of RPTP_L (Figure 3A,B).

In contrast, our model predicts that the loss of RPTP_L enhances the migrating abilities of mesenchymal cells induced by the presence of TGFB (Figure 3C). Fixing TGFB to 1 in the presence of RPTP_1 leads to a unique M1 mesenchymal phenotype, reminiscent to the increased mobile phenotypes acquired by epithelial cells treated with TGFB [59]. However, the model shifted to the M2 mesenchymal phenotype in the absence of RPTP_L (Figure 3C)

Moreover, the model anticipates that RPTP_L cooperates with a stiff ECM to support the gain of the H3 hybrid phenotype (Figure 3D). Stiffening of the ECM, simulated by fixing ECM to 1, is capable of generating EMT phenotypes, with the gain of M3 mesenchymal and H3 hybrid phenotypes, each in about 25% of the cases. The remaining 50% of the simulation runs, showing a maintenance of the E1 phenotype. Consistent with this prediction, growing the non-transformed human mammary epithelial cell line MCF10A on a stiff matrix is sufficient to induce a partial EMT phenotype [60]. However, the H3 hybrid phenotype could no longer be reached when fixing RPTP_L to 0.

Furthermore, the loss of RPTP_L could also synergise with growth factors (Figure 3E,F). In the presence of RPTP_L, fixing HGF or EGF to 1 does not permit the model to leave the E1 phenotype. In contrast, in the absence of RPTP_L, the model reaches the H2 hybrid phenotype in over 70% of the cases, as well as the M2 mesenchymal phenotype with a lower probability. Surprisingly, while our model predicts that the presence of HGF is not sufficient to induce EMT phenotypes (in the presence of RPTP_L), untransformed epithelial cells treated with HGF have been shown to undergo EMT [61,62]. These data were gathered using cells grown on plastic, which are far stiffer than even the stiffest living tissue. We thus simulated a stiff ECM in the presence of HGF or EGF. With ECM and HGF or EGF set to 1, together with RPTP_L, the model reaches the H3 and M3 phenotypes, reminiscent of the effect of a stiff ECM alone (Figure 3, compare E and F with D). Strikingly, the loss of RPTP_L also cooperates with growth factors and ECM stiffening, as the model reaches a unique M3 mesenchymal phenotype in the presence of a stiff ECM and HGF or EGF when RPTP_L is set up to 0. Taken together, our model suggests that microenvironment inputs are, for many of them, unable or poorly potent to induce EMT phenotypes on their own, but while in synergy with RPTP_L, they become powerful EMT triggers.

### 3.2. NOTCH Cooperates with Inputs from the Tumour Microenvironment in EMT

In agreement with experimental observations [57,63], the model indicates that fixing DELTA to 1 (Appendix A), or simulating a NOTCH GoF mutation (NOTCH E1) (Figure 4), does not permit the model to leave the E1 phenotype. Yet, GoF mutations in NOTCH have been reported in patients with solid cancers [64]. As NOTCH has been shown to play a critical role in EMT [65], we searched for accomplices of NOTCH in EMT dynamics, by performing systematic analyses of the model behaviour upon switching inputs one by one in the presence of a NOTCH E1 mutation (Appendix A).

Starting from an E1 phenotype, with NOTCH E1, and fixing DELTA or WNT to 1, together with RPTP_L, does not permit to leave the E1 phenotype (Appendix A). NOTCH does not cooperate with ROS in EMT. Indeed, with ROS fixed to 1, the model reaches identical phenotypes without or with the NOTCH E1 mutation (Figure 4A). In contrast, the model is no longer able to attain a H2 hybrid phenotype when IL6 is set to 1 in the presence of NOTCH E1 (Figure 4B). Similar to ROS, NOTCH does not cooperate with TGFB (Figure 4C). Noteworthy, with the NOTCH E1 mutation, the model loses the ability to reach a H3 hybrid phenotype when ECM is fixed to 1 (Figure 4D, compare with Figure 4D). However, when HGF or EGF are set to 1 (together with RPTP_L), NOTCH E1 does not permit to leave the E1 phenotype (Figure 4E,F). Thus, all input-dependent hybrid phenotypes are hindered by the presence of a NOTCH GoF mutation.

To confirm these observations, we tested if NOTCH E1 could also eliminate the H2 hybrid phenotypes reached when HGF or EGF are set to 1 in the absence of RPTP_L. Indeed, the model could only reach the M2 mesenchymal phenotype in these simulations. As the loss of RPTP_L alone does not affect the outcome of a NOTCH GoF mutation, this indicates that the effect of NOTCH E1 in promoting a mesenchymal phenotype is dependent on both: the loss of RPTP L and the presence of HGF or EGF. Taken together, the model predicts that a NOTCH GoF mutation cooperates with microenvironment inputs to convert hybrid phenotypes into mesenchymal ones.

### 3.3. NOTCH Cooperates with Secondary Mutations in EMT

To further extend our search for cooperative interactions between NOTCH and other model variables in EMT, we performed a systematic analysis of the phenotypes compatible with the 80 LoF and GoF single mutants and by the 78 combinations of double mutants involving a NOTCH GoF mutation (Appendix A). This analysis was carried out through a Python script available upon request. This script interacts with GINsim (more precisely with the Java toolkit bioLQM [66]) to define model perturbations, and to get the stable states of perturbed models. Cooperating mutations should display the acquisition of phenotypes not observed in either single mutants or the loss of a phenotype observed in both single mutants.

The model predicts that the E1 phenotype is compatible with single GoF mutations in NOTCH (NOTCH E1), RAF (RAF E1), MEK (MEK E1) and ERK (ERK E1). However, this phenotype is lost in double mutants involving NOTCH E1, suggesting that NOTCH and RAS signalling cooperate to prevent the maintenance of an E1 phenotype (Table 2 and Appendix A). In agreement with these predictions, expressing a constitutive active form of NOTCH or low levels of oncogenic RAS is not sufficient to transform the immortalized HMLE breast cell line, while co-expressing both induces efficient colonies in soft agar and tumours in nice mice [63].

In addition, NOTCH E1 and STAT3 E1 synergise to impede the E1 phenotype (Table 2). STAT3 and NOTCH signalling are known to crosstalk in EMT by activating each other [67,68]. Yet, our model proposes that additional cooperative mechanisms are in play between NOTCH and STAT3 to trigger EMT.

In addition, we found five mutants cooperating with NOTCH E1 to support hybrid phenotypes. Single NOTCH E1 and PI3K LoF mutants (PI3K KO) are each compatible with the gain of the H3 hybrid phenotype. In contrast, this phenotype is lost in the double mutant (Table 2 and S2). Consistent with these predictions, neither a constitutive NOTCH activation, nor a PI3K overactivation caused by a null mutation in its upstream negative regulator Pten, induce distal metastases in a mouse model for prostate cancer. In contrast, distal metastases are observed in Pten-null mice overactivating NOTCH [57,69].

In addition, the model predicts that NOTCH cooperates with YAP_TAZ and WNT signalling to generate hybrid phenotypes, as the single mutants NOTCH E1, LATS E1, YAP_TAZ KO, DVL KO and CK1 KO give rise to the H1, H2 and H3 phenotypes, while these phenotypes are absent in the double mutants involving NOTCH E1. Hybrid states have been proposed to endow cells with cancer stem cell (CSC) properties [65], suggesting that the cooperation between NOTCH and YAP_TAZ or WNT signalling supports the acquisition of stemness properties. Accordingly, NOTCH, YAP_TAZ, and WNT signalling have all been implicated in endorsing CSC properties [70]. NOTCH could cooperate with YAP or WNT signalling by synergistically regulating common target genes, as previously reported in other contexts [71,72,73].

## 4. Conclusions and Prospects

Treatment regiments using small molecule inhibitors, mostly targeting intracellular kinases, have improved the clinical outcomes in patients with cancer [74]. Still, clinical trials in oncology have one of the lowest success rates among all diseases [75]. Among the factors that explain this lack of success is our poor knowledge of a multitude of interconnected cell-autonomous and non-cell-autonomous determinants, which, upon changing conditions (acquisition of mutations, alterations in the TME) can cooperate or antagonise each other to generate cancer phenotypes. In this context, computational models can efficiently overcome the limitations of reductionist approaches in cancer research by capturing tumour complexity and by generating powerful hypotheses.

Logical models of interaction networks have proven to help identify individual factors that may work jointly in generating cancer phenotypes (see Section 2.2). Our search for cooperativity in EMT, between RPTP_L, NOTCH, and other inputs from the TME (Figure 5A), or between NOTCH and other model variables (Figure 5B), serves as an illustration of the predictive power of logical models. Indeed, we could recapitulate synergistic behaviours already reported experimentally and propose novel cooperative mechanisms that remain to be confirmed. Computational models can identify “gene-chameleons” behaving in an opposite way during oncogenesis. An example could be RPTPs. Our model anticipates that the loss of RPTP_L synergises with growth factors, ECM stiffening, and TGFB to promote EMT (Figure 3). However, we have also reported a critical role of RPTP in supporting an H3 hybrid phenotype, typical of collective cell migration behaviour [44]. This mode of migration appears more efficient in establishing metastases than individual migrating cancer cells, and this correlates with worse patient outcomes [76]. Hence RPTP could fulfil a cancer-promoting function in this context. Accordingly, RPTPs have been classified as both tumour suppressors and oncogenes [77].

In silico models may also reveal that, depending on the identity of its accomplice, an oncogene can give rise to different phenotypes. Our model simulations predict that a NOTCH GoF mutation synergises with IL6, ECM stiffening, or growth factors in the absence of RPTP_L to convert hybrid phenotypes into mesenchymal ones (Figure 4). In contrast, NOTCH E1 cooperates with PI3K, YAP_TAZ, or WNT signalling to induce hybrid phenotypes (Table 2). Hence, treatments with single agents targeting NOTCH signalling could be detrimental for patients, as in the presence of high levels of IL6, growth factors, or a stiff ECM, NOTCH inhibitors could trigger hybrid phenotypes, therefore favouring the acquisition of cells with CSCs properties.

In addition, logical models are suitable to seek translational goals by proposing timely combinations of targets for therapeutic benefit, before the transition to the next tumoral stage. Generic or tissue specific models can be built to encompass key regulatory circuits and biomarkers involved in carcinogenesis and core cellular decisions (e.g., cell division, survival, adhesion properties, differentiation). These models could be implemented in clinical settings, for personalised medicine, as predictors of effective therapeutic regimens [78]. Genomic, transcriptomic, and proteomic datasets are being obtained from patient tumour biopsies. These data can be used to train logical models in order to recapitulate tumour behaviours and to predict molecular targets, reverting detrimental phenotypes to non-cancer ones. These predictions can subsequently be tested on organoids, derived from the same tumour material, before clinical practice (Figure 6).

There exists a variety of mathematical formalisms to build dynamical models of cellular networks, from more quantitative systems of differential equations to qualitative, discrete logical models. Each approach presents its own benefits and limitations, and the choice of a modelling framework greatly depends on the size of the system to be studied, on available data, on questions to be addressed, etc. Here, we have focused on the logical formalism, a popular framework for the analysis of large networks. It goes beyond the scope of this review to discuss advantages and limitations of this modelling framework or to compare it with other approaches. For such a discussion, we refer to e.g., [24,79].

Finally, it is worth noting that the EMT model, as other cellular models mentioned in this review, consider cells in a fixed environment. To appropriately account for the interplay between cells and their environment, and to assess the dynamics of populations of cells, it is necessary to rely on multi-scale modelling approaches. Software tools have been developed to enable the definition of such models, still in a discrete framework. For example, EpiLog defines logical models over hexagonal grids of communicating cells [80], and PhysiBoSS combines an agent-based framework with a Boolean modelling of the intra-cellular networks [81].

## Figures and Tables

**Figure 1 ijms-22-04897-f001:**
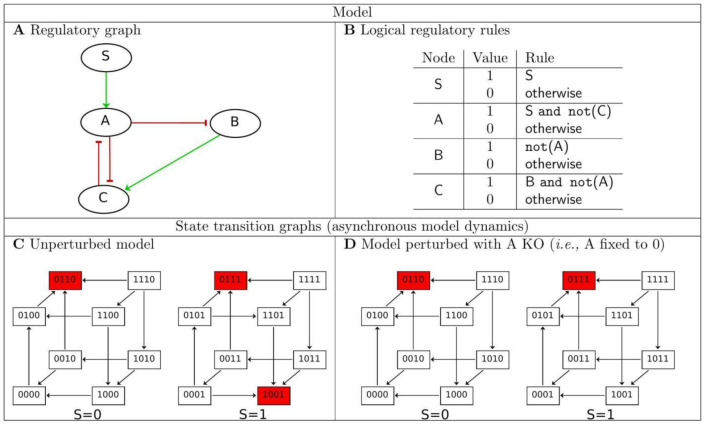
A simple logical (Boolean) model and its dynamics. (**A**) The regulatory graph, where nodes represent model components (three internal and one input nodes), green arrowhead edges denote activations, and red blunt edges denote inhibition; (**B**) Each node value is defined by a logical rule, i.e., the input is maintained constant at its current value, A is activated (A = 1) if S, its activator, is present and C, its inhibitor is absent, otherwise it is de-activated (A = 0), etc.; (**C**) The state transition graph representing the dynamics of the unperturbed model; each node represents a model state with the values of the components [A,B,C,S]; arrowhead edges represent transitions, and trajectories are defined by successions of transitions; when S = 0, there is a single stable state (0110), all trajectories converge to it; when S = 1, there are 2 stable states (0111) and (1001); starting from state (0001), both stable states are reachable, (0111) with probability 0.2514, and (1001) with probability 0.7486; (**D**) The state transition graph when the value of A is blocked to zero, which mirrors a LoF (KO) of A, showing that when S = 1, a stable state is lost.

**Figure 2 ijms-22-04897-f002:**
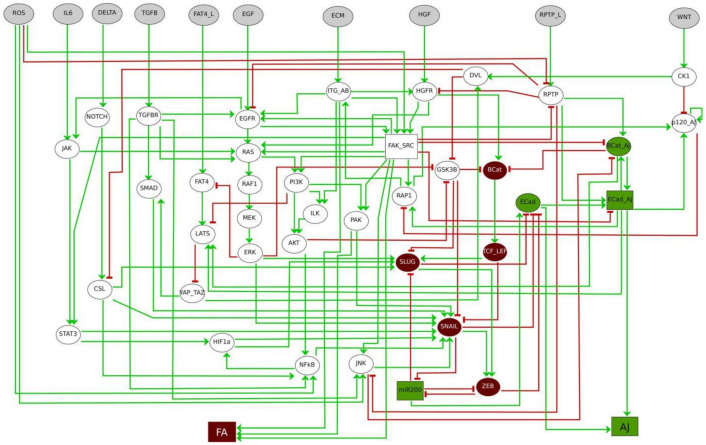
Regulatory network of cell adhesion properties controlled by the micro-environment during EMT. Inputs from the TME are denoted in grey, epithelial markers are indicated in green, whereas mesenchymal markers are in dark brown. Oval nodes denote Boolean components, rectangular ones denote multi-valued components. Green arrows denote activatory interactions, red blunt arrows denote inhibitory interactions. Two model read-outs provide the states of Adherens Junction (AJ) assembly and of Focal Adhesion (FA) recycling.

**Figure 3 ijms-22-04897-f003:**
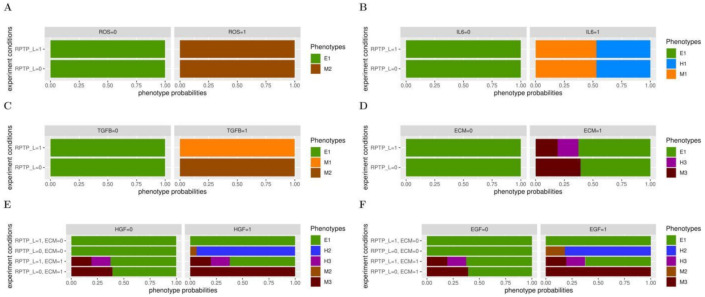
The EMT logical model predicts cooperation between RPTP_L and other microenvironment signals. Probabilities of the reachable phenotypes starting from an E1 phenotype, when switching single or multiple microenvironmental signals. (**A**–**D**) ROS (**A**) or IL6 (**B**) or TGFB (**C**) or ECM (**D**) is set up to 0 or 1, in the presence (=1) or absence (=0) of RPTP_L. (**E**,**F**) HGF (**E**) or EGF (**F**) is set up to 0 or 1, in the presence (=1) or absence (=0) of RPTP_L and of a stiff ECM. In each condition, inputs that are not indicated are fixed to 0.

**Figure 4 ijms-22-04897-f004:**
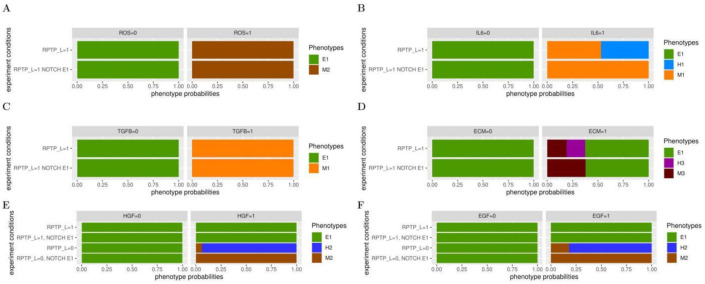
The EMT logical model predicts cooperation between a NOTCH GoF mutation and microenvironment signals. Probabilities of the reachable phenotypes, starting from an E1 phenotype, in the presence or absence of a NOTCH GoF mutation (NOTCH E1) when switching microenvironmental signals. (**A**–**D**) ROS (**A**) or IL6 (**B**) or TGFB (**C**) or ECM (**D**) is set up to 0 or 1, in the presence (=1) or absence (=0) of NOTCH E1. (**E**,**F**) HGF (**E**) or EGF (**F**) is set up to 0 or 1, in the presence (=1) or absence (=0) of NOTCH E1 and of RPTP_L. In each condition, inputs that are not indicated are fixed to 0.

**Figure 5 ijms-22-04897-f005:**
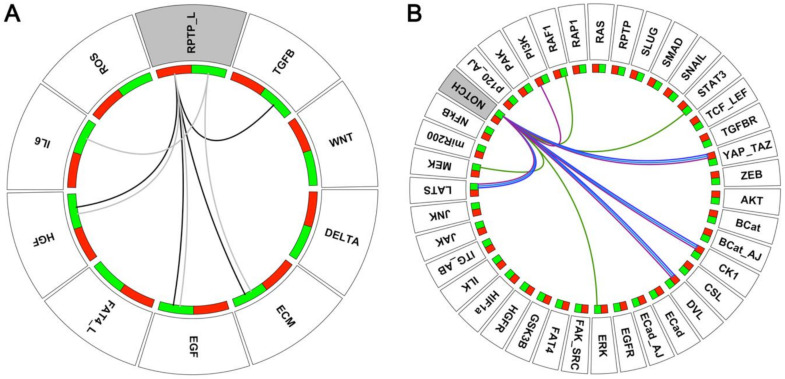
Diagrams summarising cooperative interactions predicted by our EMT model (**A**) Cooperative interactions between RPTP_L and other inputs from the TME in the absence of a NOTCH Gof mutation (black edges) or in the presence of a NOTCH GoF mutation (grey edges) using reachability analysis starting from the E1 phenotype (see Appendix A). Coloured rectangles associated with each input provide the value of the input, green for the value 1 (active) and red for 0 (non-active). All remaining inputs are set to 0. (**B**) Cooperative interactions between a NOTCH GoF mutation (NOTCH E1) and other internal variables of the model, by systematic analysis of the phenotype compatible with the different conditions. Coloured edges connecting two variables indicate the phenotypes lost in double mutants but observed in both single mutants: green edges for the epithelial E1 phenotype, light blue edges for the hybrid H1 phenotype; darker blue edges for the hybrid H2 phenotype and magenta edges for the hybrid H3 phenotype. Coloured rectangles associated with each variable provide the type of mutations: green for a GoF mutation, red for a LoF mutation.

**Figure 6 ijms-22-04897-f006:**
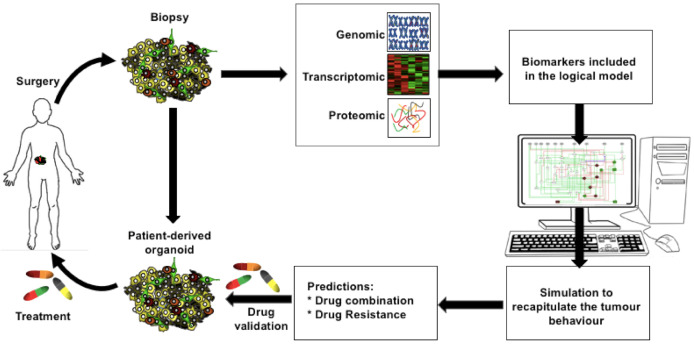
Scheme to establish personalised chemotherapeutic drug regiments using logical modelling. A tumour biopsy is processed to gather genomic, transcriptomic, and proteomic datasets and to establish cultured organoids. A logical model, including the biomarkers identified in the tumour datasets, is trained to recapitulate the tumour state. Starting from this state, systematic model perturbations allow predictions of drug combinations, which can revert the tumour state to a non-tumour state and overcome possible drug resistance mechanisms. Predicted drug combinations are validated in patient-derived organoid before safe clinical practice.

**Table 1 ijms-22-04897-t001:** The attractors of the logical EMT model identify eight phenotypes characterised by the values of the read-out components AJ and FA. The first column indicates the phenotypes and their classification into: epithelial, hybrid, unknown, or mesenchymal based on the values of the read-outs AJ and FA. The second and third columns indicate the values of the read-outs AJ and FA, respectively. The last column describes each phenotype.

Phenotypes	AdherensJunctions(AJs)	Focal Adhesions(FAs)	Description
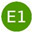	Epithelial	2	0	AJ assembly due to ECad-BCat interaction at the membrane
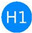	Hybrid	2	1	AJ assembly due to ECad-BCat interaction at the membrane, combined with a weak ability to recycle FAs
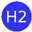	1	2	Failure to assemble AJs while maintaining ECad expression, combined with an intermediate ability to recycle FAs
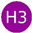	2	3	AJ assembly due to Ecad-Bcat interaction at the membrane, combined with a high ability to recycle FAs
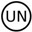	Unknown	0	0	Lack of either epithelial or mesenchymal markers, undefined phenotype
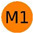	Mesenchymal	0	1	Failure to assemble AJs and to express epithelial markers, combined with the expression of mesenchymal markers and a weak ability to recycle FAs
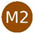	0	2	Failure to assemble AJs and to express epithelial markers, combined with the expression of mesenchymal markers and an intermediate ability to recycle FAs
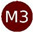	0	3	Failure to assemble AJs and to express epithelial markers, combined with the expression of mesenchymal markers and a high ability to recycle FAs

**Table 2 ijms-22-04897-t002:** NOTCH GoF mutations cooperate with RAS, STAT3, PI3K, Hippo, and WNT signalling in EMT. The first column indicates each phenotype and their classification into the epithelial, hybrid, unknown, or mesenchymal based on the values of the read-outs AJ and FA. The second and third columns indicate the values of the read-outs AJ and FA, respectively. The remaining 20 columns correspond to phenotypes of the unperturbed model or when a single GoF (E1), a LoF (KO) mutation or double mutations involving a NOTCH GoF mutation is introduced. Tick symbol indicates the presence of the corresponding phenotype, whereas grey cells indicate its absence.

Phenotypes	AJ stability	FA recycling	Unperturbed model	NOTCH E1	RAF E1	NOTCH E1 RAF E1	MEK E1	MEK E1 NOTCH E1	ERK E1	ERK E1 NOTCH E1	STAT3 E1	STAT3 E1 NOTCH E1	PI3K KO	PI3K KO NOTCH E1	LATS E1	LATS E1 NOTCH E1	YAP_TAZ KO	YAP_TAZ KO NOTCH E1	DVL KO	DVL KO NOTCH E1	CK1 KO	CK1 KO NOTCH E1
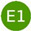	Epithelial	2	0	✓	✓	✓		✓		✓		✓		✓	✓	✓	✓	✓	✓	✓	✓	✓	✓
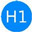	Hybrid	2	1	✓	✓	✓	✓	✓	✓	✓	✓	✓	✓			✓		✓		✓		✓	
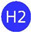	1	2	✓	✓	✓	✓	✓	✓	✓	✓	✓	✓	✓	✓	✓		✓		✓		✓	
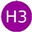	2	3	✓	✓	✓	✓	✓	✓	✓	✓	✓	✓	✓		✓		✓		✓		✓	
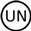	Unknown	0	0	✓	✓	✓	✓	✓	✓	✓	✓	✓	✓	✓	✓								
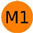	Mesenchymal	0	1	✓	✓	✓	✓	✓	✓	✓	✓	✓	✓			✓	✓	✓	✓	✓	✓	✓	✓
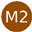	0	2	✓	✓	✓	✓	✓	✓	✓	✓	✓	✓	✓	✓	✓	✓	✓	✓	✓	✓	✓	✓
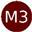	0	3	✓	✓	✓	✓	✓	✓	✓	✓	✓	✓	✓	✓	✓	✓	✓	✓	✓	✓	✓	✓

## Data Availability

The EMT model, as published in [44], was made available in the GINsim model repository (http://ginsim.org/model/EMT_Selvaggio_etal). It was also deposited in BioModels [82] and assigned the identifier MODEL2004040001.

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
