# Peer review of "In Silico Logical Modelling to Uncover Cooperative Interactions in Cancer"

_ijms, 2021, doi:10.3390/ijms22094897_

Round 1
Reviewer 1 Report
- Introduction
This section is generally good. It may be useful to define some less well known molecular terms at the start (particularly as many readers may be computational scientists). For example, ‘epigenetic lesions’ are not defined.
Line 37: The comment “(related to cancer-promoting genes)” should mention both cancer promoting genes (i.e. protooncogenes) and tumour suppressor genes.
Line 44-46: You say that the number of mutations causing the majority of cancers varies from 3-12. But then immediately state that the minimum number is 2. It’s a minor difference, but the phrasing could be improved so the sentences are more consistent with each other.
Line 60: Could you explain better what you mean by “In addition, cooperative cell behaviours have been reported between genetically related cancer cells to influence their aggressiveness”? Does this refer to sibling cells within a clone, or cells that arose independently with the same mutation? Or does ‘genetically related’ mean cells that began with the same mutational origin, but subsequently diverged by acquiring different mutations? How do they cooperate?
Line 77: This line misses a point about the nature of tumour evolution. I would say: “Thus, cooperative interactions are likely highly dynamic reflecting the ongoing process of tumour evolution under the influence of a continually shifting genetic and epigenetic landscape, and altered signaling from the TME. “
Line 92: what do you mean by “..physical mutagenesis”? Do you mean ionizing radiation? If you mean genetic engineering that is covered below in transgenesis, HR and CRISPR.
Line 98: the section on Drosophila should include important caveats. A key one being lack of a closed circulatory system so the role of the vasculature on cancer cell dissemination can’t be studied. Also, common forms of human cancer such as breast and prostate that are driven by circulating factors such as hormones (e.g. steroids) can’t be modelled.
- Computational modelling approaches to pinpoint cooperative interactions in oncogenesis
Line 116: Do you mean non-linear in the strict sense of dose-response relationships? Or are you referring to the intersection of orthogonal signalling events? If the former, this seems very significant when confined to binary values for node states as in most logical models. How do you handle factors/pathways bell shaped response curves in such models?
Line 125: It isn’t clear what species you are talking about – do you mean chemical species? If so, this is the first time you mention chemical entities and properties of reactions (e.g. kinetics). Prior to this all focus was on genes and signalling pathways. If you are speaking of chemicals that may be important regulators in cancer, then this concept needs to be introduced first. If by kinetics you are in fact referring to ligand/receptor binding affinity as a determinant of signalling activity, this important concept hasn’t been introduced yet either.
Overall, I would say handling quantitative (analogue) input in logical models is a limitation that needs to be better addressed, either in the Introduction or Discussion. It isn’t always sufficient to know that the ligand and receptor are present/not present, cancer is very often driven by a quantitative change in the signal, which may in turn be due to a change in ligand level (e.g. steroid driven breast and prostate cancers, excess bile acid as a driver of CRC).
The fundamental value of logical models is not in question, just their limitations in modelling real world complexity, particularly if each node is assigned a binary state. The argument was made that logical modelling can overcome reductionism because you are not limited by the number of components that can be modelled. But actually these models just seem to reduce complexity in a different way. Rather than removing components, they instead reduce the number of states a component can have. My interpretation is that if more realistic analogue (e.g. continuous range) values are used for the nodes, the computational demand of any problem involving multiple nodes will become unsupportable. I would appreciate the authors comments on this issue. Including whether they believe such concerns can be partly obviated by the fact that some signalling pathways show threshold effects.
Relevant to the above point, in the discussion of modelling ‘omics data in section 2.2., this data is by nature quantitative and continuous, yet there is little explained about exactly how this type of data is used to inform logical models.
On line 221, the TCGA is mentioned without explaining what this database is and its relevance to extrapolating data from cell-based studies to clinical observations.
On line 239, what exactly are ‘hybrid EMT phenotypes’?
On line 249, please clarify the statement: “From this analysis, a synergy emerged between NOTCH GoF and p53 LoF, which together lead the sole metastatic phenotype..” Does this indicate that this combination gives 100% probability of this phenotype?
On line 269, how were the confirmatory studies done (other than real time growth assays) e.g. was siRNA knockdown used in pairwise combinations?
The section around line 275 that discusses the relative predictive power of modelling gene expression changes vs genetic changes (mutation) could be expanded/emphasized a bit, since the modelling of transcriptomic data seems to be the main theme of much of this section.
Line 288: The sentence “To confront logical models to clinical data, Béal et al. recently proposed an approach integrating mutation data, copy number alterations, transcriptomic and proteomic data to models [31].” Isn’t easy to parse. What models are you taking about and how exactly can clinical data be integrated? This might be a good place to discuss the different types/levels of ‘omics data available, and draw distinctions between data from cell based studies in which specific perturbations are possible, vs ‘omics data from actual patient tumours (such as in TCGA).
- Logical modelling predicts cooperatives signals governing phenotypes amid the EMT continuum
In general, this whole section may benefit if more of the information from Ref 41 was included that explained how the experimental validation was done. It would make the robustness of the modelling approach in predicting biologically relevant outcomes more apparent. Also, I find the use of the term WT in Table 2 slightly confusing. I understand these to be cancer cells without specific mutations/perturbations that can naturally traverse the EMT continuum and hence all 8 states are achievable (indicated by the tick). However, my concern is that to a causal reader, WT may suggest normal (non-cancer) epithelial cells. A brief explanation of the terminology at the start of section 3 may be useful. My final general comment on this section is that NOTCH signalling is notoriously complex and ON/OFF states are something of an over simplification as different ligand and receptor modifications transduce altered forms of the signal that effect selective responses. This should be considered as part of interpreting/extrapolating the model outcomes.
Line 320: Is there a reference for the concept of: “…a hybrid H3 phenotype, reminiscent to the one displayed by cancer cells that migrate collectively”. Is this hybrid phenotype something that has been seen elsewhere in biology, or did it just emerge from your modelling?
Line 344: I would restate this as “The underlying biological assumptions underlying our model were that in the context of a non-tumorigenic TME….”
Also, in this section (around line 348) I would clarify that node state 0 = OFF and 1 = ON.
- Discussion
This is generally good, but again, there seem to be caveats to the real world extrapolation of logical models that reduce complexity by assigning the nodes binary states (or at least very few states). This should be discussed either in the Introduction or Discussion.
Typos/grammatical changes
(there are many, I tried to catch most)
Line 20: “…logical computational models combined with experimental validation,.. “
Line 35: “ … thousands of genes…”
Line 55: “…help protect against..”
Line 57: “…tumour cells supporting their survival..”
Line 66: “ … emergence of cancer phenotypes…”
Line 73: “Moreover, many genes can act as either oncogenes or tumour suppressors in different experimental settings. The contradictory functions of these “chameleon” genes depends on…”
Line 77: “Thus, cooperative interactions are likely highly dynamic as tumours progress and evolve under the influence of a continually shifting genetic and epigenetic landscape, and signals from the TME.”
Line 81: “ Providing evidence of cooperativity between two molecular alterations or between two cell populations requires that the phenotypic…”
Line 84: “… have been valuable instruments to determine…”
Line 92: “..generated via a variety of methods..”
Line 99: “..which can be genetically modified and maintained with comparatively basic training and infrastructure ..”
Line 100: “…in core cancer relevant genes..”
Line 101: “ .. multiple mutated transgenes..”
Line 103: “..tools developed in Drosophila allow assessment of how..”
Line 112: “..approaches to characterize the interplay of multiple regulatory components in cancer complexity..”
Page 120: “…high throughput ‘omics technologies (e.g., genomics, epigenomics, transcriptomics, metabolomics, proteomics, and others) …”
Pg 123: “In this review..”
Ph 143: “..by the sequences of states,..”
Line 192: “..consists of comparing..”
Line 195: “..attaching a (fixed) value..”
Line 198: ‘..Much modelling work focuses on networks..”
Line 234: “..Model prediction and experimental validation in hepatocellular carcinoma cell lines indicates that the concomitant activation of WNT and SHH signalling are required..”
Line 248: “..probability of obtaining a metastatic phenotype..”
Line 301: “Logical modelling predicts cooperative signals …”
Line 315: “…among which are 3 hybrid (H) ones that co-express epithelial and mesenchymal markers…”
Line 324: “..Thus, our model can uncover..”
Line 383: “..could not be reached at all when..”
Line 394: “..does not permit cells to leave..”
Line 403: “..reminiscent of the effect..”
Line 408: “…they become powerful..”
Line 415: “..does not permit cells to leave..”
Line 518: “..Logical models of interaction networks have proven valuable to help identify individual factors that may work jointly..”
Line 521: “..behaving in opposite ways during oncogenesis. An example could be the RPTP family of genes.”
Line 557: “..allow prediction of drug combinations..”
Line
There is a typo in Table 1 (epithelial or mesenchymal markers)
Author Response
- Introduction
Reviewer: This section is generally good. It may be useful to define some less well known molecular terms at the start (particularly as many readers may be computational scientists). For example, ‘epigenetic lesions’ are not defined.
Response: We have revised this section to clarify molecular terms for an audience that might not be very familiar with oncogenesis. Epigenetic lesions is not appropriate. We have rephrased this paragraph.
Reviewer: Line 37: The comment “(related to cancer-promoting genes)” should mention both cancer promoting genes (i.e. protooncogenes) and tumour suppressor genes.
Response: We have altered the text as follow: “Some of these alterations are drivers of oncogenesis. They have been classified as oncogenes or tumour suppressors depending on whether they are constitutively activated by gain-of-function (GoF) mutations or inactivated by loss-of function (LoF) mutations, respectively. In contrast, other alterations are likely non-oncogenic with no selective advantage.”
Reviewer: Line 44-46: You say that the number of mutations causing the majority of cancers varies from 3-12. But then immediately state that the minimum number is 2. It’s a minor difference, but the phrasing could be improved so the sentences are more consistent with each other.
Response: The number of mutations required to cause oncogenesis in human has been estimated from 3 to 12. Yet, in cell lines and Drosophila, 2 cooperative mutations can lead to phenotypes akin to cancer development. To justify the disparity in term of minimum number of mutations required to trigger oncogenesis, we have specified that estimates of the number of mutations causing the large majority of malignant cancer vary from 3 to 12 in human.
Reviewer: Line 60: Could you explain better what you mean by “In addition, cooperative cell behaviours have been reported between genetically related cancer cells to influence their aggressiveness”? Does this refer to sibling cells within a clone, or cells that arose independently with the same mutation? Or does ‘genetically related’ mean cells that began with the same mutational origin, but subsequently diverged by acquiring different mutations? How do they cooperate?
Response: The two last sentences of the paragraph on clonal cooperativity were indeed confusing. We have rephrased these two sentences to point out in the first sentence cooperativity between clones of cells carrying distinct mutations, and in the second sentence cooperativity between clones of cells that have the same mutational origin but are likely distinct in term of epigenetic modifications as follows: “Cooperative behaviours are also observed between clones of tumour cells carrying different genetic lesions, including evidence supporting the requirement of interactions between distinct clones for tumour initiation [12,13]. In addition, clonal cooperativity have been reported between cancer cells with no discernible genetic differences to influence their aggressiveness, suggesting a key role of epigenetic modifications [14,15].”
We have also replaced the review from Barbara L. Parsons on multiclonal tumour origin by two research articles showing cooperativity between clones of cells carrying distinct genetic lesions [12,13].
We have also included a sentence highlighting that cooperative mechanisms likely involve paracrine signalling as follows: “Mechanisms of cooperativity in play likely involve paracrine signalling between clones of cells [12,14].”
Reviewer: Line 77: This line misses a point about the nature of tumour evolution. I would say: “Thus, cooperative interactions are likely highly dynamic reflecting the ongoing process of tumour evolution under the influence of a continually shifting genetic and epigenetic landscape, and altered signaling from the TME. “
Response: To include the notion of tumour evolution, we propose the following sentence: “Thus, cooperative interactions are likely highly dynamic during tumour evolution, owing to the emergence of novel mutations, epigenetic alterations and signals present in the TME.”
Reviewer: Line 92: what do you mean by “..physical mutagenesis”? Do you mean ionizing radiation? If you mean genetic engineering that is covered below in transgenesis, HR and CRISPR.
Response: The physical mutagenesis mentioned here refers to gamma rays, X rays, UV light, and particle radiation. We have included this information between brackets in the text for clarification.
Reviewer: Line 98: the section on Drosophila should include important caveats. A key one being lack of a closed circulatory system so the role of the vasculature on cancer cell dissemination can’t be studied. Also, common forms of human cancer such as breast and prostate that are driven by circulating factors such as hormones (e.g. steroids) can’t be modelled.
Response: We have included caveats on the use of Drosophila in cancer research as follows: “Yet, as the fly lacks similarities in telomere and telomeric maintenance strategies compared to human and does not possess fibroblasts, adaptive immune system and vasculature, the role of these players can hardly be evaluated in Drosophila cancer models.”
This statement leaves the possibility that cancer-associated processes with no straightforward counterpart in human could still be study in the fly. For example, the regulation of the interconnected tubular tracheal system in the fly which, like the mammalian vascular system, spreads oxygen throughout the body, is significantly analogue to that of the mammalian vascular tree. Like angiogenesis, tracheogenesis is induced by hypoxic stress in tumours through HIF1α/Sima-dependent activation of signalling pathways, consequently resulting in increasing access to oxygen (Grifoni, D.; Sollazzo, M.; Fontana, E.; Froldi, F.; Pession, A. Multiple strategies of oxygen supply in Drosophila malignancies identify tracheogenesis as a novel cancer hallmark. Sci. Rep. 2015, 5, 1–11, doi:10.1038/srep09061).
.
- Computational modelling approaches to pinpoint cooperative interactions in oncogenesis
Reviewer: Line 116: Do you mean non-linear in the strict sense of dose-response relationships? Or are you referring to the intersection of orthogonal signalling events? If the former, this seems very significant when confined to binary values for node states as in most logical models. How do you handle factors/pathways bell shaped response curves in such models?
Response: In this specific sentence, we meant non-linear to underscore that signalling pathways are not simple linear cascades. We have clarified this by changing the sentence as follows: “Computational models can simulate the complex dynamics driven by intricate signalling pathways with feedbacks and cross-talks. As such, they are effective tools to decipher cooperative mechanisms in the context of cancer biology. “
The reviewer is right in referring to the non-linearity of most regulatory mechanisms. The logical formalism handles it by abstracting e.g. Hill functions and bell shaped functions by step functions, possible considering distinct threshold values.
Reviewer: Line 125: It isn’t clear what species you are talking about – do you mean chemical species? If so, this is the first time you mention chemical entities and properties of reactions (e.g. kinetics). Prior to this all focus was on genes and signalling pathways. If you are speaking of chemicals that may be important regulators in cancer, then this concept needs to be introduced first. If by kinetics you are in fact referring to ligand/receptor binding affinity as a determinant of signalling activity, this important concept hasn’t been introduced yet either.
Response: By species we refer to all the system variables (miRNA, kinase, phosphatase, transcription factors). They are characterised by their concentrations and a series of kinetic parameters specific to their role (e.g. binding affinities, turnover number). However, to avoid confusion, we have rephrased this part as follows: “Unlike differential equations, logical models do not require precise values for the concentrations of molecular species, gene expression levels, or kinetic constants that are mostly lacking in the literature.”
Reviewer: Overall, I would say handling quantitative (analogue) input in logical models is a limitation that needs to be better addressed, either in the Introduction or Discussion. It isn’t always sufficient to know that the ligand and receptor are present/not present, cancer is very often driven by a quantitative change in the signal, which may in turn be due to a change in ligand level (e.g. steroid driven breast and prostate cancers, excess bile acid as a driver of CRC).
The fundamental value of logical models is not in question, just their limitations in modelling real world complexity, particularly if each node is assigned a binary state. The argument was made that logical modelling can overcome reductionism because you are not limited by the number of components that can be modelled. But actually these models just seem to reduce complexity in a different way. Rather than removing components, they instead reduce the number of states a component can have. My interpretation is that if more realistic analogue (e.g. continuous range) values are used for the nodes, the computational demand of any problem involving multiple nodes will become unsupportable. I would appreciate the authors comments on this issue. Including whether they believe such concerns can be partly obviated by the fact that some signalling pathways show threshold effects.
Response: The value of a logical model resides in the possibility of developing a parameter free model. Continuous or discrete kinetic models require a large amount of information that are not always available or even measurable. As an example, a Hill equation requires n and Kd (Hill coefficient and dissociation constant), and the more the values are uncertain, the more the model will be qualitative.
In any case, we agree with the reviewer that a logical model cannot achieve the same results as a continuous model. For small systems, in the literature they are plenty of continuous models disclosing behaviours that could not be revealed with the level of abstraction inherent of logical models.
However, this discussion on the pros and cons of different modelling approaches is not the purpose of the present paper. We have added a short paragraph in the discussion with appropriate references.
Reviewer: Relevant to the above point, in the discussion of modelling ‘omics data in section 2.2., this data is by nature quantitative and continuous, yet there is little explained about exactly how this type of data is used to inform logical models.
Response: We have added a short paragraph with some explanation as follows: ”For such a contextualisation of logical models, as illustrated in Figure 1 of [33], data need to be appropriately processed through functional inference (in the case of discrete genomic data such as mutations or copy numbers) and through discretisation and normalisation (in the case of e.g., expression data).”
Reviewer: On line 221, the TCGA is mentioned without explaining what this database is and its relevance to extrapolating data from cell-based studies to clinical observations.
Response: We now explain the purpose of the TCGA database and how datasets from this resource where used to validate the model prediction as follows: “These predictions were supported by datasets from the TCGA database, which provides genomic data from over 20.000 primary cancer and matched normal samples. Among the five tumours carrying mutations in FGFR3 and p21CIP, four also display an homozygous deletion of the CDKN2A gene.”
Reviewer: On line 239, what exactly are ‘hybrid EMT phenotypes’?
Response: We have now defined hybrid EMT phenotypes when referring for the first time to EMT on line 316 as follows: “Another hallmark of tumour cells that has motivated a substantial number of logical models is the Epithelial to Mesenchymal Transition (EMT), a representative example of cancer cell plasticity [38]. This process not only involves a switch between the two extreme phenotypes, epithelial and mesenchymal, but also the transition to a spectrum of incomplete EMT phenotypes. These hybrid phenotypes, which co-express epithelial and mesenchymal markers, have been proposed to provide pluripotent abilities to cancer cells, resistance to chemotherapeutic drugs and increased aggressive potential [39].”
Reviewer: On line 249, please clarify the statement: “From this analysis, a synergy emerged between NOTCH GoF and p53 LoF, which together lead the sole metastatic phenotype..” Does this indicate that this combination gives 100% probability of this phenotype?
Response: We have clarified this sentence as follows: “From this analysis, a synergy emerged between NOTCH GoF and p53 LoF, which together lead to metastatic phenotype in 100% of the runs,…”
Reviewer: On line 269, how were the confirmatory studies done (other than real time growth assays) e.g. was siRNA knockdown used in pairwise combinations?
Response: The study by Flobak et al. only provides real time growth assays to validate their predictions on synergistic inhibition on cell growth. They used chemical inhibitors for involved components in single or combinatorial formulations. We have now clarified this point as follows: “Searching for pairs of inhibitors of cell growth, the model predicted five synergistic inhibitions involving seven components. Among those, four cooperative inhibitions were confirmed experimentally, by comparing the effect on cell growth of treating the AGS gastric cancer cell line with chemical inhibitors of the seven proteins, in single or combinatorial formulations, using cell growth real-time assays.”
Reviewer: The section around line 275 that discusses the relative predictive power of modelling gene expression changes vs genetic changes (mutation) could be expanded/emphasized a bit, since the modelling of transcriptomic data seems to be the main theme of much of this section.
Response: Transcriptomic data have indeed been used to confront logical models, but more recently phosphoproteomic data have also been considered (e.g. ref 31). It is not our intention to discuss/compare “the relative predictive power of modelling gene expression changes vs genetic changes (mutation)“. In any case, due to the diversity of mechanisms that can be handled in logical models (transcriptional regulation, post-transcriptional regulation, signalling, etc), the predictive power of the model will relate to the modeled mechanisms.
Reviewer: Line 288: The sentence “To confront logical models to clinical data, Béal et al. recently proposed an approach integrating mutation data, copy number alterations, transcriptomic and proteomic data to models [31].” Isn’t easy to parse. What models are you taking about and how exactly can clinical data be integrated? This might be a good place to discuss the different types/levels of ‘omics data available, and draw distinctions between data from cell based studies in which specific perturbations are possible, vs ‘omics data from actual patient tumours (such as in TCGA).
Response: We believe that, although highly relevant, a discussion about the different types of data, and their comparative value in the context of cancer modelling studies goes beyond the scope of our paper. Nevertheless, we have rephrased the mentioned sentence as follows: “To personalise logical models, Béal et al. recently proposed an approach integrating mutation data, copy number alterations (CNA), transcriptomic and proteomic data to models [33]. The authors illustrated the value of their framework using a generic model of cancer signalling pathways [48] and breast cancer data from the METABRIC project, including RNA expression data, mutation profiles, CNA and clinical data [49,50]. As illustrated in Figure 1 of [33], data need to be appropriately processed through functional inference (in the case of discrete genomic data such as mutations or CNA) and through discretisation and normalisation (in the case of e.g., expression data).”
- Logical modelling predicts cooperatives signals governing phenotypes amid the EMT continuum
Reviewer: In general, this whole section may benefit if more of the information from Ref 41 was included that explained how the experimental validation was done. It would make the robustness of the modelling approach in predicting biologically relevant outcomes more apparent. Also, I find the use of the term WT in Table 2 slightly confusing. I understand these to be cancer cells without specific mutations/perturbations that can naturally traverse the EMT continuum and hence all 8 states are achievable (indicated by the tick). However, my concern is that to a causal reader, WT may suggest normal (non-cancer) epithelial cells. A brief explanation of the terminology at the start of section 3 may be useful. My final general comment on this section is that NOTCH signalling is notoriously complex and ON/OFF states are something of an over simplification as different ligand and receptor modifications transduce altered forms of the signal that effect selective responses. This should be considered as part of interpreting/extrapolating the model outcomes.
Response: We have included additional information on experimental data validating the model prediction as follow: “Model predictions indicate that the FAK-SRC complex cooperates with a stiff ECM to upregulate SNAIL and to induce a full mesenchymal phenotype. Experimental validations using the MCF10A and MDCK cell lines with conditional SRC activation, grown on collagen gels of different Young´s moduli, which mirror a soft ECM surrounding normal mammary epithelial cells or a stiffer matrix reported for stroma adjacent to transformed cells, confirmed that SNAIL expression was significantly higher in SRC overactivating cells grown on stiff gels compared to those plated on soft one. Moreover, while SRC overactivating cells grown on stiff gels were isolated and accumulated poorly ECad at the cell membrane, those plated on soft gels maintained epithelial features characterized by the presence of cell–cell contacts at ‘tip-like’ junctions and membrane-associated ECad. In addition, model simulations revealed that FAK-SRC cooperates with RPTP, which mediates homophilic cell-cell adhesion, to gain a hybrid H3 phenotype, reminiscent to the one displayed by cancer cells that migrate collectively [39]. According to this prediction, forcing PTPR-kappa expression in MCF10A with conditional SRC activation using the (CRISPR)-based activation system significantly could restore cell aggregation.”
We have changed the term WT in Table 2 and now use “unperturbed model”.
Reviewer: Line 320: Is there a reference for the concept of: “…a hybrid H3 phenotype, reminiscent to the one displayed by cancer cells that migrate collectively”. Is this hybrid phenotype something that has been seen elsewhere in biology, or did it just emerge from your modelling?
Response: We have included a reference, which reviews in silico, in vitro, in vivo and clinical evidence for the existence of hybrid EMT phenotype(s) in multiple carcinomas facilitating collective cancer cell migration [39].
Reviewer: Line 344: I would restate this as “The biological assumptions underlying our model were that in the context of a non-tumorigenic TME….”
Response: We have replaced “Our reasoning was that, in the context of a non-tumorigenic TME,…” by “The underlying biological assumptions of our model were that, in the context of a non-tumorigenic TME,…”
Reviewer: Also, in this section (around line 348) I would clarify that node state 0 = OFF and 1 = ON.
Response: we believe that the meaning of node values 0 and 1 has been appropriately explained in section 2.1 “(1 for active, 0 for non-active)”. Nevertheless, we have modified Figure 1 panel D, clarifying the meaning of “A KO”.
- Discussion
Reviewer: This is generally good, but again, there seem to be caveats to the real world extrapolation of logical models that reduce complexity by assigning the nodes binary states (or at least very few states). This should be discussed either in the Introduction or Discussion.
Response: we consider that a discussion on the pros and cons of different modelling frameworks would go far beyond the scope of the present paper. Our goal here was to illustrate the appropriateness of the logical approach to disclose cooperative mechanisms in cancer. However, to address the reviewer concern, we have included a few sentences in the discussion as follows: “There exists a variety of mathematical formalisms to build dynamical models of cellular networks, from more quantitative systems of differential equations to qualitative, discrete logical models. Each approach presents its own benefits and limitations, and the choice of a modelling framework greatly depends on the size of the system to be studied, on available data, on questions to be addressed, etc. Here, we have focused on the logical formalism, a popular framework for the analysis of large networks. It goes beyond the scope of this review to discuss advantages and limitations of this modelling framework, or to compare it with other approaches. For such a discussion, we refer to e.g., [24,80].”
Typos/grammatical changes
(there are many, I tried to catch most)
Reviewer: Line 20: “…logical computational models combined with experimental validation,.. “
Response: We have replaced “to” by “with”
Reviewer: Line 35: “ … thousands of genes…”
Response: We have replaced “…thousands genes…” by “…thousands of genes…”
Reviewer: Line 55: “…help protect against..”
Response: We have replaced “…help protecting against…” by “…help protect against…”
Reviewer: Line 57: “…tumour cells supporting their survival..”
Response: We have replaced “…tumour cells for their survival…” by “…tumour cells supporting their survival…”
Reviewer: Line 66: “ … emergence of cancer phenotypes…”
Response: We have replaced “…emergence of the cancer phenotypes…” by “….emergence of cancer phenotypes…”
Reviewer: Line 73: “Moreover, many genes can act as either oncogenes or tumour suppressors in different experimental settings. The contradictory functions of these “chameleon” genes depends on…”
Response: We have replaced “Moreover, many genes can act as both oncogenes and tumour suppressors in different experimental settings. The antagonistic function of these “chameleons” genes depends on…” by “Moreover, many genes can act as either oncogenes or tumour suppressor genes in different experimental settings. The opposite effects of these “chameleons” genes depends on…”
Reviewer: Line 77: “Thus, cooperative interactions are likely highly dynamic as tumours progress and evolve under the influence of a continually shifting genetic and epigenetic landscape, and signals from the TME.”
Response: We have rephrased the sentence as follows: Thus, cooperative interactions are likely highly dynamic during tumour evolution, owing to the emergence of novel mutations, epigenetic alterations and signals present in the TME.
Reviewer: Line 81: “ Providing evidence of cooperativity between two molecular alterations or between two cell populations requires that the phenotypic…”
Response: We took out the “s” in “evidence” and replaced “implies” by “requires”
Reviewer: Line 84: “… have been valuable instruments to determine…”
Response: We have replace “….have been instrumental to determine…” by “…have been valuable models to determine…”
Reviewer: Line 92: “..generated via a variety of methods..”
Response: We have replaced “…have been generated thanks to a variety of methods…” by “…have been generated via a variety of methods…”
Reviewer: Line 99: “..which can be genetically modified and maintained with comparatively basic training and infrastructure ..”
Response: We have replaced the sentence: “….which requires basic training and infrastructure.” by “….which can be genetically modified and maintained with comparatively basic training and infrastructure.”
Reviewer: Line 100: “…in core cancer relevant genes..”
Response: We have replaced “…in cancer relevant genes…” by “…in core cancer relevant genes…”
Reviewer: Line 101: “ .. multiple mutated transgenes..”
Response: We removed the “s” to multiple
Reviewer: Line 103: “..tools developed in Drosophila allow assessment of how..”
Response: We have replaced “…tools developed in Drosophila allow to assess how…” by “…tools developed in Drosophila allow assessment of how…”
Reviewer: Line 112: “..approaches to characterize the interplay of multiple regulatory components in cancer complexity..”
Response: We have replaced “…approaches in revealing the role of multiple components into the cancer complexity.” by “…approaches to characterise the interplay of multiple regulatory components in cancer complexity.”
Reviewer: Page 120: “…high throughput ‘omics technologies (e.g., genomics, epigenomics, transcriptomics, metabolomics, proteomics, and others) …”
Response: We have replaced “…high throughput omics technologies (e.g., genomics, transcriptomics, metabolomics or proteomics, and others).” by “…high throughput omics technologies (e.g., genomics, epigenomics, transcriptomics, metabolomics, proteomics, and others).”
Reviewer: Pg 123: “In this review..”
Response: We have replaced …” in this article…” by “…in this review..”
Reviewer: Ph 143: “..by the sequences of states,..”
Response: We have replaced “…by sequences of states…” by “….by the sequences of states…”
Reviewer: Line 192: “..consists of comparing..”
Response: We have replaced “…consists in comparing…” by “…consists of comparing…”
Reviewer: Line 195: “..attaching a (fixed) value..”
Response: we have changed as follows: “Altering a model component entails fixing the value of the corresponding variable,...”
Reviewer: Line 198: ‘..Much modelling work focuses on networks..”
Response: We have modified the sentence as follows (not to convey the wrong fact that a majority of models focused on proliferation and death: “Several modelling work…. “
Reviewer: Line 234: “..Model prediction and experimental validation in hepatocellular carcinoma cell lines indicates that the concomitant activation of WNT and SHH signalling are required..”
Response: We removed the “s” to “validation”
Reviewer: Line 248: “..probability of obtaining a metastatic phenotype..”
Response: We have replaced “…. probability to get a metastatic phenotype.” by “….probability of reaching a metastatic phenotype.”
Reviewer: Line 301: “Logical modelling predicts cooperative signals …”
Response: We removed the “s” to “cooperative”
Reviewer: Line 315: “…among which are 3 hybrid (H) ones that co-express epithelial and mesenchymal markers…”
Response: We have replaced “…3 hybrid (H) ones, co-expressing epithelial and mesenchymal markers” by “…”…among which are 3 hybrid (H) ones that co-express epithelial and mesenchymal markers.”
Reviewer: Line 324: “..Thus, our model can uncover..”
Response: We have replaced “Thus, our model permits to uncover…” by “Thus, our model can uncover…”
Reviewer: Line 383: “..could not be reached at all when..”
Response: We have replaced “….could not be reached anymore…” by “…could no longer be reached…”
Reviewer: Line 394: “..does not permit cells to leave..”
Response: We have replaced “…does not permit to leave…..” by “…does not permit the model to leave…”
Reviewer: Line 403: “..reminiscent of the effect..”
Response: We have replaced “…reminiscent to the effect..” by “…reminiscent of the effect..”
Reviewer: Line 408: “…they become powerful..”
Response: We have replaced “…they turn into powerful EMT triggers” by “…they become powerful EMT triggers.”
Reviewer: Line 415: “..does not permit cells to leave..”
Response: We have replace “…does not permit to leave….” by “ does not permit the model to leave..”
Reviewer: Line 518: “..Logical models of interaction networks have proven valuable to help identify individual factors that may work jointly..”
Response: We have replaced “Logical models of involved interaction networks have proven valuable for the academic pursuit to single out likely factors working jointly to generate cancer phenotypes…” by “Logical models of interaction networks have proven valuable for the academic pursuit to help identify individual factors that may work jointly in generating cancer phenotypes….”
Reviewer: Line 521: “..behaving in opposite ways during oncogenesis. An example could be the RPTP family of genes.”
Response: We have replaced “…behaving in opposite way during oncogenesis. One example of such a gene could be RPTPs.” by “behaving in opposite way during oncogenesis. An example could be RPTPs.”
Reviewer: Line 557: “..allow prediction of drug combinations..”
Response: We have replaced “…allow to predict drug combinations…” by “….allow predictions of drug combinations…”
Reviewer: There is a typo in Table 1 (epithelial or mesenchymal markers)
Response: Table 1 has been corrected
Reviewer 2 Report
Selvaggio et al review the use of logical modelling to identify cooperative interactions in the context of cancer and their possible implication for therapeutic strategies. The authors provide a short reminder of the logical modeling, and summarize successful applications of this approach to identify cooperativity between signalling components (e.g. the interplay between FGFR3, PI3K and CDKN2A). They then describe in more detail their previous work (published last year in Cancer Research) on the Epithelial–Mesenchymal Transition (EMT). In this work they devised a comprehensive logical model that recapitulates the phenotypic diversity observed in the EMT continuum (i.e. epithelial, mesenchymal and hybrid states). Simulations of the model under various microenvironmental conditions highlight several cooperative mechanisms. In particular, this study revealed the interaction of RPTP_L (or the loss of RPTP_L) with several input components, including growth factors. This study is here further extended and lead to additional predictions such as the NOTCH-RAS, the NOTCH-STAT, or the NOTCH-YAP/WNT cross-talks and their consequences for the resulting phenotypes.
The paper is well written and nicely illustrates the potential of logical modeling in deciphering the complex interplay between signalling pathways and their consequences for the development of cancer.
I only have a few minor suggestions:
1) It is not very clear how the probabilities of the reachability of steady states are calculated. For example, in the toy model (Figure 1), the authors say that "starting from state [0001], both stable states are reachable, [0111] with probability 0.2514, and [1001] with probability 0.7486". A brief explanation on how these probability have been obtained would be welcome.
2) A figure showing the complete model for the EMT would be useful. Such a scheme was provided in the Cancer Research paper (fig. 1). If it could be reproduced here it would help the reader to better appreciate the size and complexity of the system. If the cooperative interactions could be highlighted on such a scheme this would be even more informative.
3) As I understand, in these simulations, each cell is considered as independent and in a fixed environment (which determines the presence/absence of the inputs). In reality, depending on their state, the cells can alter their microenvironment through the secretion of signalling factors. The reachabilities/probabilities computed in this study may thus be affected if we would consider such (local) inter-cellular coupling in an heterogeneous population of cells. A comment about this assumption/limitation could be added in the conclusions.
Author Response
Reviewer: Selvaggio et al review the use of logical modelling to identify cooperative interactions in the context of cancer and their possible implication for therapeutic strategies. The authors provide a short reminder of the logical modeling, and summarize successful applications of this approach to identify cooperativity between signalling components (e.g. the interplay between FGFR3, PI3K and CDKN2A). They then describe in more detail their previous work (published last year in Cancer Research) on the Epithelial–Mesenchymal Transition (EMT). In this work they devised a comprehensive logical model that recapitulates the phenotypic diversity observed in the EMT continuum (i.e. epithelial, mesenchymal and hybrid states). Simulations of the model under various microenvironmental conditions highlight several cooperative mechanisms. In particular, this study revealed the interaction of RPTP_L (or the loss of RPTP_L) with several input components, including growth factors. This study is here further extended and lead to additional predictions such as the NOTCH-RAS, the NOTCH-STAT, or the NOTCH-YAP/WNT cross-talks and their consequences for the resulting phenotypes.
The paper is well written and nicely illustrates the potential of logical modeling in deciphering the complex interplay between signalling pathways and their consequences for the development of cancer.
I only have a few minor suggestions:
1) It is not very clear how the probabilities of the reachability of steady states are calculated. For example, in the toy model (Figure 1), the authors say that "starting from state [0001], both stable states are reachable, [0111] with probability 0.2514, and [1001] with probability 0.7486". A brief explanation on how these probability have been obtained would be welcome.
Response: we have added a short explanation as follows: To do so, a Monte Carlo approach, consisting of N (asynchronous) simulation runs starting from an initial state and counting the number R of runs ending in e.g., the stable state [0111] provides R/N as a good approximation of this probability [25].
Reviewer: 2) A figure showing the complete model for the EMT would be useful. Such a scheme was provided in the Cancer Research paper (fig. 1). If it could be reproduced here it would help the reader to better appreciate the size and complexity of the system. If the cooperative interactions could be highlighted on such a scheme this would be even more informative.
Response: We have included a new figure with the model network (now Figure 2). As the model loses visibility when highlighting cooperative interactions uncovered in this work, we propose a new figure (Figure 5), summarising the cooperative interactions we observed between RPTP_L and other inputs from the TME in the absence or presence of a NOTCH Gof mutation (Figure 5A) or between NOTCH and other variables of the model (Figure 5B), considering exclusively dual cooperation between inputs from the TME We refer to this figure in the conclusion and prospect section.
Reviewer: 3) As I understand, in these simulations, each cell is considered as independent and in a fixed environment (which determines the presence/absence of the inputs). In reality, depending on their state, the cells can alter their microenvironment through the secretion of signalling factors. The reachbilities/probabilities computed in this study may thus be affected if we would consider such (local) inter-cellular coupling in an heterogeneous population of cells. A comment about this assumption/limitation could be added in the conclusions.
Response: We thank the reviewer for this comment. We have added a few sentences of this issue at the end of the discussion as follows:
Finally., it is worth noting that the EMT model as other cellular models mentioned in this review amount to consider cells in a fixed environment. To appropriately account for the interplay between cells and their environment and to assess the dynamics of populations of cells, it is necessary to rely on multi-scale modelling approaches. Software tools have been developed to enable the definition of such models, still in a discrete framework- For example, EpiLog defines logical models over hexagonal grids of communicating cells [81] and PhysiBoss combines an agent-based framework with a Boolean modelling of the intra-cellular networks [82].